# Economical Production of Phenazine-1-carboxylic Acid from Glycerol by *Pseudomonas chlororaphis* Using Cost-Effective Minimal Medium

**DOI:** 10.3390/biology12101292

**Published:** 2023-09-27

**Authors:** Yu-Xuan Li, Sheng-Jie Yue, Yi-Fan Zheng, Peng Huang, Yan-Fang Nie, Xiang-Rui Hao, Hong-Yan Zhang, Wei Wang, Hong-Bo Hu, Xue-Hong Zhang

**Affiliations:** 1State Key Laboratory of Microbial Metabolism, School of Life Sciences and Biotechnology, Shanghai Jiao Tong University, Shanghai 200240, China; yuxuanli@sjtu.edu.cn (Y.-X.L.); sjyue@sjtu.edu.cn (S.-J.Y.); hpflb-14@alumni.sjtu.edu.cn (P.H.); yanfangnie18@163.com (Y.-F.N.);; 2Shanghai Nong Le Biological Products Company Limited (NLBP), Shanghai 200240, China; 3Shanghai Nongle Joint R&D Center on Biopesticides and Biofertilizers, Shanghai Jiao Tong University, Shanghai 200240, China

**Keywords:** phenazine-1-carboxylic acid, *Pseudomonas chlororaphis*, minimal medium, fermentation optimization, genetic engineering, glycerol utilization

## Abstract

**Simple Summary:**

Phenazine compounds are widely used in agricultural control and the medicine industry. Synthesizing phenazine compounds through microbial fermentation often requires a complex and relatively expensive culture medium, which greatly limits the large-scale industrial production of phenazine compounds by fermentation. Therefore, a cost-effective minimal medium (GDM) for the efficient synthesis of phenazine compounds by *Pseudomonas chlororaphis* was developed in this work. Using the GDM, the production of phenazine compounds by *P. chlororaphis* reached 1073.5 mg/L, which was 1.3 times that achieved using a complex medium, while the cost of the GDM was only 10% that of a complex medium (e.g., the KB medium). To improve the utilization of glycerol in the GDM, the glycerol metabolic pathway was enhanced by the genetic engineering of the strain, and the titer of phenazine-1-carboxylic acid of the engineered strain was further improved, which is a promising process for industrial production.

**Abstract:**

Phenazine compounds are widely used in agricultural control and the medicine industry due to their high inhibitory activity against pathogens and antitumor activity. The green and sustainable method of synthesizing phenazine compounds through microbial fermentation often requires a complex culture medium containing tryptone and yeast extract, and its cost is relatively high, which greatly limits the large-scale industrial production of phenazine compounds by fermentation. The aim of this study was to develop a cost-effective minimal medium for the efficient synthesis of phenazine compounds by *Pseudomonas chlororaphis*. Through testing the minimum medium commonly used by *Pseudomonas*, an ME medium for *P. chlororaphis* with a high production of phenazine compounds was obtained. Then, the components of the ME medium and the other medium were compared and replaced to verify the beneficial promoting effect of Fe^2+^ and NH_4_^+^ on phenazine compounds. A cost-effective general defined medium (GDM) using glycerol as the sole carbon source was obtained by optimizing the composition of the ME medium. Using the GDM, the production of phenazine compounds by *P. chlororaphis* reached 1073.5 mg/L, which was 1.3 times that achieved using a complex medium, while the cost of the GDM was only 10% that of a complex medium (e.g., the KB medium). Finally, by engineering the glycerol metabolic pathway, the titer of phenazine-1-carboxylic acid reached the highest level achieved using a minimum medium so far. This work demonstrates how we systematically analyzed and optimized the composition of the medium and integrated a metabolic engineering method to obtain the most cost-effective fermentation strategy.

## 1. Introduction

Due to the depletion of fossil resources, climate change, and other environmental problems caused by existing unsustainable industries, the global public’s interest in more sustainable production processes is rapidly increasing [1,2,3]. However, some industries still rely on chemical synthesis for manufacturing the required products, which should be reformed in order to achieve better sustainability. The chemical synthesis of phenazine compounds, which exhibit high inhibitory activity against pathogens and antitumor activity, is a typical case, requiring strict reaction conditions (e.g., aqueous alkali, high temperature) and possibly producing some toxic by-products [4,5]. The microbial synthesis of phenazine compounds (e.g., phenazine-1-carboxylic acid, PCA; phenazine-1-carboxamide, PCN; and 2-hydroxyphenazine, 2-OH-PHZ) is an alternative to chemical synthesis, attracting more and more attention due to its mild reaction conditions and green and sustainable process. The mechanism of the PCA biosynthesis pathway has been elucidated clearly. Phenazines are derived from chorismate, the end product of the shikimate pathway. The gene cluster *phzABCDEFG* codes the enzymes that facilitate phenazine biosynthesis in Pseudomonas. In addition, one glutamine needs to be consumed in the synthesis of phenazine, since the first step is the conversion of chorismate to 2-amino-4-deoxychorismic acid (ADIC) by PhzE with the consumption of one amino group of glutamine.

The microorganisms that can synthesize phenazine compounds mainly include *Pseudomonas* [6], *Streptomyces* [7], and *Methanosarcina* [8]. The fermentation time of *Streptomyces* is much longer than that of *Pseudomonas*; thus, its fermentation cycle is longer, which is not conducive to industrial production. Additionally, the culture conditions of *Methanosarcina* are harsh, and genetic manipulation is difficult [9]; therefore, it is difficult to use metabolic engineering for high-yield transformation. Moreover, *P. aeruginosa*, which can synthesize phenazine compounds, is an opportunistic pathogen and not suitable for industrial fermentation to produce phenazine compounds [10]. In addition, *P. chlororaphis* also synthesizes phenazine compounds by the fermentation method [11]. Furthermore, this strain possesses two excellent characteristics: (1) it is a non-pathogenic plant-growth-promoting rhizobacteria [12]; (2) it can synthesize phenazine compounds using glycerol as a carbon source [11]. Glycerol is an inevitable by-product of the biodiesel industry, and 1 kg crude glycerol is generated per 10 kg of biodiesel production [13]. With the acceptance of biodiesel as a better alternative to fossil fuels worldwide and the continuous expansion in the scale of the relevant enterprises, surplus glycerol has become an important economic problem in the biodiesel industry [1,14]. For example, Zhu et al. proposed *B. amyloliquefaciens* for the efficient synthesis of poly-γ-glutamic acid (γ-PGA) from crude glycerol as a single carbon source through a systematic metabolic engineering strategy, resulting in a γ-PGA titer of 26.4 g/L [15]. Therefore, this is a perfect scheme to maximize the production of phenazine compounds by *P. chlororaphis* using glycerol as a substrate.

*P. chlororaphis* is mainly cultured in a complex medium containing tryptone or yeast to effectively synthesize phenazine compounds, and the cost of the culture medium is relatively high [16]. However, when *P. chlororaphis* is cultured in a low-cost minimal medium containing glycerol, the titer of phenazine compounds is very low [17]. In this study, we screened and compared some minimal medium commonly used by *Pseudomonas* and optimized the trace elements and nitrogen source. From this, we generalized a cost-effective general defined medium (GDM) that can be used for the efficient synthesis of phenazine compounds by *P. chlororaphis*. In general, the cost of the GDM was 1/10 that of the complex medium for the same PCA production. Finally, engineering the glycerol metabolic pathway further increased the yield of the strain by 27.4%, resulting in the more cost-effective synthesis of phenazine compounds by glycerol. We also tested the phenazine synthesis ability of the engineered strain in the minimal medium with crude glycerol as the sole carbon source, and the result proved that the engineered strain had a certain tolerance to the inhibitors in crude glycerol. This result indicated the potential of *P. chlororaphis* as a cell factory for use with economical and green carbon sources in GDM, such as lignocellulose hydrolysates and sugarcane waste.

## 2. Materials and Methods

### 2.1. Bacterial Strains and Basic Culture Conditions

*P. chlororaphis* GP72-ANO (wild-type GP72 with the inactivation of the *rpeA* and *phzO* genes) and its derivatives [11] (ANOE: strain ANO with empty plasmid; ANOF: strain ANO with *glpF* gene overexpression plasmid; ANOK: strain ANO with *glpK* gene overexpression plasmid; ANOFK: strain ANO with *glpF* and *glpK* gene co-overexpression plasmid; ANOFK1: strain ANOΔ*mgsA* with *glpF* and *glpK* gene co-overexpression plasmid; ANOFK2: strain ANOΔ*mgsA*Δ*glpR* with *glpF* and *glpK* gene co-overexpression plasmid) were activated and pre-cultured ingrown in King’s medium B (glycerol 15 mL, tryptone 20 g, MgSO_4_ 0.732 g, K_2_HPO_4_ 0.514 g/L) at 28 °C.

The fermentation performance of *P. chlororaphis* GP72-ANO was tested in minimal medium (AB medium: 2 g/L (NH_4_)_2_SO_4_, 6 g/L Na_2_HPO_4_, 3 g/L KH_2_PO_4_, 3 g/L NaCl, 1 mL 0.1 M CaCl_2_, 1 mL1.0 M MgCl_2_, 1 mL 0.003 M FeCl_3_; MM medium: 1.62 g/L NH_4_Cl, 1.25 g/L NaH_2_PO_4_·2H_2_O, 0.2 g/L MgSO_4_, 2.21 g/L K_2_HPO_4_, 15 mg/L Na_2_EDTA·2H_2_O, 4.5 mg/L CoCl_2_·6H_2_O, 0.3 mg/L ZnSO_4_·7H_2_O, 1 mg/L MnCl_2_·4H_2_O, 1 mg/L H_3_BO_3_, 2.5 mg/L CaCl_2_, 0.4 mg/L Na_2_MoO_4_·2H_2_O, 3 mg/L FeSO_4_·7H_2_O, 0.3 mg/L CuSO_4_·5H_2_O; M9 medium: 6 g/L Na_2_HPO_4_, 3 g/L KH_2_PO_4_, 1.4 g/L (NH_4_)_2_SO_4_, 0.5 g/L NaCl, 0.2 g/L MgSO_4_·7H_2_O, 300 mg/L HBO_3_, 50 mg/L ZnCl_2_, 30 mg/L MnCl_2_·4H_2_O, 200 mg/L CoCl_2_, 10 mg/L CuCl_2_·2H_2_O, 20 mg/L NiCl_2_·6H_2_O, 30 mg/L NaMoO_4_·2H_2_O; ME medium: 3.3 g/L (NH_4_)_2_HPO_4_, 5.8 g/L K_2_HPO_4_, 3.7 g/L KH_2_PO_4_, 0.12 g/L MgSO_4_, 2.78 mg/L FeSO_4_·7H_2_O, 1.98 mg/L MnCl_2_·4H_2_O, 2.38 mg/L CoCl_2_·6H_2_O, 1.67 mg/L CaCl_2_·2H_2_O, 0.17 mg/L CuCl_2_·2H_2_O, 0.29 mg/L ZnSO_4_·7H_2_O) supplemented with 20 g/L glycerol as the sole carbon source. The substrate crude glycerol was purchased from Nantong Dayao Chemical Co., Ltd. (Nantong, China), and its glycerol content was 75%.

For fermentation, strain GP72-ANO and its derivatives stored at −80 °C in a freezer were activated twice at 28 °C overnight on a King’s medium B plate (glycerol 15 mL, tryptone 20 g, MgSO_4_ 0.732 g, K_2_HPO_4_ 0.514 g/L, 2.0% agar). A single colony from the LB plate was then inoculated into 5 mL LB medium at 28 °C overnight and washed with the corresponding minimal medium to serve as a seed culture. Shaking flask fermentations were carried out in 60 mL of a corresponding minimal medium at 28 °C and 200 rpm. For overexpression strains, the medium were supplemented with kanamycin antibiotics (Kn, 50 μg/mL) and Ampicillin antibiotics (Ap, 100 μg/mL), and 1 mM IPTG (isopropyl-β-D-thiogalactopyranoside) was added to the medium at 6 h to induce gene overexpression.

### 2.2. Determination of Cell Growth and Phenazine Compound Titer

Cell growth was measured using an ultraviolet spectrophotometer by monitoring the absorbance value of the culture at a wave of 600 nm. To avoid the interference of phenazine compounds on absorbance, centrifugation and resuspension with clean water were necessary before detection. The fermentation broth of *P. chlororaphis* (0.4 mL) was acidified to pH 2.0 with 6 M HCl and centrifuged at 10,000 rpm for 5 min using an Eppendorf Minispin centrifuge. Then, the supernatant was mixed with 3.6 mL ethyl acetate and shaken vigorously for 5 min. The extraction supernatant (0.4 mL) was transferred to a new EP tube for air drying. After thorough drying, 1 mL acetonitrile was added for dissolution. Then, the resulting residues were filtered through a 0.22 μm filter for high-performance liquid chromatography (Agilent 1260, CA, USA) detection. We used an Agilent Eclipse XDB-C18 reversed-phase column (4.6 mm × 250 mm, Agilent, Santa Clara, CA, USA) eluted with acetonitrile and 0.1% formic acid (60:40, *v*/*v*) at a scanning wavelength of 254 nm and a flow rate of 1.0 mL/min.

### 2.3. Statistical Analysis

All of the results were averaged and are depicted as the mean ± standard deviation from at least triplicate independent experiments.

## 3. Results

### 3.1. Comparison of Fermentation Performance of P. chlororaphis in Different Minimal Medium

Low-cost minimal medium (M9, MM, AB, and ME) were used to culture *P. chlororaphis* GP72-ANO (PCA-producing strain), *P. chlororaphis* LX24 (2-OH-PHZ-producing strain) and *P. chlororaphis* HT66-H7 (PCN-producing strain). As shown in Figure 1a, the phenazine compound titers of the three strains in the ME medium were higher than those of the other three medium. The titer of phenazine compounds in the strain GP72-ANO fermentation broth cultured in ME medium was the highest, reaching 440.2 mg/L (Figure 1a), and the OD_600_ of its broth was 18.5 (Figure 1b). In our previous study, the titer of the GP72-ANO fermentation broth in the commonly used KB medium (a complex medium) was around 800 mg/L [11]. All of the tested strains produced a low titer in the commonly used M9 and MM medium (Figure 1a). Meanwhile, the maximum biomass of the three strains tested in the two medium (i.e., M9 and MM) was only about 20% that in the ME medium (Figure 1b). Herein, based on this result, we selected a minimal ME medium that was more suitable for the synthesis of phenazine compounds by *P. chlororaphis* GP72-ANO (the strain with the highest yield of phenazine compounds in the minimal medium used in this test). Next, we wanted to explore which components in the ME medium were conducive to the synthesis of phenazines.

### 3.2. Effect of Trace Elements in ME Medium on PCA Titer by P. chlororaphis GP72-ANO

The components in the medium with a concentration at the mg/L level were defined as trace elements in this study, and we first assumed that trace elements in the ME medium played a dominant role. The trace elements in the four medium are shown in Figure 2a, and the ME medium contained CoCl_2_, ZnSO_4_, MnCl_2_, and CuSO_4_, as did the MM and AB medium. In order to test whether the trace elements in the ME medium promoted the efficient synthesis of PCA by *P. chlororaphis* GP72-ANO, the trace elements of the AB, MM, and M9 medium were replaced with the trace elements of the ME medium, resulting in the AB-ME, MM-ME, and M9-ME medium. The PCA titer of the strain GP72-ANO broth in the AB-ME medium reached 370.8 mg/L, which was 445.5% and 84.2% that in the AB and ME medium (Figure 2b). The PCA titers of strain GP72-ANO in MM-ME and M9-ME were 1.6- and 18.9-fold that in the M9 and MM medium, respectively (Figure 2b). These results suggested that the trace elements in the ME medium may have promoted the PCA synthesis of strain GP72-ANO, which also confirmed our hypothesis. Then, the trace elements of the ME medium were replaced with those of the MM, M9, and AB medium, resulting in the ME-MM, ME-M9, and ME-AB medium. The PCA titer and cell growth of strain GP72-ANO in the ME-MM medium were similar to those in the ME medium (Figure 2c). Both the ME medium and MM medium contained CoCl_2_, ZnSO_4_, MnCl_2_, and CuSO_4_, as well as CaCl_2_ and FeSO_4_ (Figure 2a). Therefore, considering that these two components did not exist in the AB and M9 medium, CaCl_2_ or FeSO_4_ may have played an important role in the PCA synthesis of strain GP72-ANO in the ME and ME-MM medium. Next, all the trace elements in the ME medium were removed, and this medium was renamed the ME(N) medium. The ME(N) medium was supplemented with the same concentration of CaCl_2_ or FeSO_4_ as in the ME medium, resulting in the ME(N) + CaCl_2_ and ME(N) + FeSO_4_ medium, respectively. As shown in Figure 2d, the PCA titers of the strain GP72-ANO broth cultured in the ME(N) medium and ME(N) + CaCl_2_ medium were 42.2 and 53.3 mg/L, which were 9.5% and 12.0% that in the ME medium, respectively. However, the PCA titer and cell growth of strain GP72-ANO cultured in the ME(N) + FeSO_4_ medium were 434.8 mg/L and 17.0 (OD_600_), which were 98.3% and 95.3% that in ME medium, respectively. This result showed that, among the trace elements in the ME medium, iron played a major role in promoting the PCA synthesis of strain GP72-ANO.

### 3.3. Effect of NH_4_^+^ and Iron Concentration on PCA Titer by P. chlororaphis GP72-ANO

In the previous chapter, although the trace elements of the AB, MM, and M9 medium were replaced with those of the ME medium (i.e., the AB-ME, MM-ME, and M9-ME medium), the titers of the strain GP72-ANO broth cultured in these medium were still lower than that of the ME medium. This result suggested that the differences in the main elements also affected the synthesis of phenazine compounds by strain GP72-ANO. We calculated that the NH_4_^+^ concentrations of the AB, MM, and M9 medium were 30, 30, and 21 mM, while that of the ME medium was 50 mM. In addition, nitrogen is also an important component of the phenazine skeleton. The phenazine synthesis pathway in *Pseudomonas*, the first step of the shikimic acid pathway, requires the consumption of an amino group of glutamine for the conversion of chorismic acid to ADIC. The synthesis of glutamine requires the participation of NH_4_^+^. Thus, we speculated that a low NH_4_^+^ concentration would have adverse effects on the synthesis of PCA. Therefore, we explored the effects of different NH_4_^+^ concentrations on the synthesis of PCA by *P. chlororaphis*. We conducted NH_4_^+^ gradient concentration fermentation in the ME culture medium, maintaining the consistency of the concentrations of other components. By controlling the amount of (NH_4_)_2_HPO_4_ addition, the concentration of NH4^+^ in the culture medium was controlled at 10, 20, 30, 50, 80, 100,120, 150, 180, and 200 mM. When the NH_4_^+^ concentration was 150 mM, the highest titer of PCA reached 758.9 mg/L (Figure 3b), although the biomass of the strain was not the best. When the concentration exceeded 150 mM, the yield no longer changed as the concentration increased.

In order to further explore the effect of the iron ion valence state and iron concentration on PCA synthesis, as mentioned in the previous chapter, strain GP72-ANO was cultured with a gradient of Fe^2+^ and Fe^3+^ supplementation in the ME(N) medium. By controlling the amount of FeSO_4_ addition, the concentration of Fe^2+^ in the culture medium was controlled at 10, 20, 40, 100, 200, and 2000 μM. The same applied when controlling the concentration of Fe^3+^. The results showed that the titer of the strain GP72-ANO broth cultured in the ME(N) medium supplemented with 20 μM Fe^3+^ was the highest, reaching 1072.6 mg/L (Figure 4b). The titer of the strain GP72-ANO broth cultured in the ME(N) medium supplemented with Fe^2+^ was 8% lower than that supplemented with the same optimal concentration of Fe^3+^ (Figure 4a,b). Our results showed that Fe^3+^ was more effective than Fe^2+^ in promoting PCA synthesis in the minimal medium. To sum up, we obtained an optimal general defined medium for phenazine synthesis, which was named the GDM (an optimal general defined medium) (9.9 g/L (NH_4_)_2_HPO_4_, 5.8 g/L K_2_HPO_4_, 3.7 g/L KH_2_PO_4_, 0.12 g/L MgSO_4_, 20 μM FeCl_3_·6H_2_O). On the basis of this culture medium, different carbon sources could be used to synthesize phenazine compounds.

### 3.4. Engineering Glycerol Metabolic Pathway for More Economical PCA Synthesis

In previous reports, the glycerol metabolic pathway was engineered to maximize the metabolic flow of glycerol to PCA synthesis in a complex medium [11]. The functions of the glycerol metabolism genes involved in our modifications are shown in Figure 5a, including *glpF* (glycerol facilitator); *glpK* (glycerol kinase); and *mgsA* (methylglyoxal synthase, in a glycerol metabolic shunt pathway). Similarly, these modification strategies could also be effective in the GDM. As shown in Figure 5b, the individual overexpression of the *glpF* gene (i.e., corresponding to strain ANOF) or *glpK* gene (i.e., corresponding to strain ANOK) showed no significant improvement effect, but the co-overexpression of the *glpF* and *glpK* genes (i.e., corresponding to strain ANOFK) increased the yield of PCA by 20.0%. Moreover, the blocking of the glycerol metabolic shunt pathway (i.e., acetone aldehyde pathway, corresponding to strain ANOFK1) increased the yield of PCA to 1288.1 mg/L, which was 27.4% higher than without engineering the glycerol metabolic pathway. In addition, we also attempted to use crude glycerol, a by-product of biodiesel, as the sole carbon source in the GDM. As shown in Figure 5c, compared with pure glycerol, strain ANOFK1 can maintain a basically consistent yield in crude glycerol. This is the highest PCA production using cheap minimum medium at present, which would greatly reduce the medium cost of PCA fermentation industry.

## 4. Discussion

Although the biosynthesis of phenazine compounds by *P. chlororaphis* has already been widely explored, the medium of all the strains are complex and relatively expensive. Thus, reducing the fermentation cost of *P. chlororaphis* is of great significance for its further development and application. In the present study, we tested several minimal medium and found that the ME medium was a relatively suitable culture medium for the biosynthesis of phenazine compounds. Meanwhile, the biomass in the ME medium was much higher than in the other minimal medium. However, the expression of phenazine gene clusters in *P. chlororaphis* is strictly regulated by quorum sensing, and the expression of phenazine gene clusters cannot be triggered when the bacterial concentration is low [18,19]. Consequently, we suspected that such a low concentration of biomass may not have been enough to induce the expression of phenazine gene clusters, and that was the reason for the low yield of phenazine compounds in the minimum medium. Although the titer of phenazine compounds in the ME medium was slightly lower than that in the KB medium, the former was far superior to the KB medium in terms of economy. The KB medium contained expensive tryptone (20 g/L), and its cost was about 10 times that of the ME medium.

To further optimize the composition of the ME medium, we substituted the trace elements of the other minimal medium for those of the ME medium. Based on the substitution experiments, we demonstrated that iron played a major role in promoting phenazine compound (PCA) synthesis. By comparing the differences in the NH_4_^+^ concentration between the four minimal medium, we speculated that NH_4_^+^ was one of the key reasons why the ME medium was superior to the other three minimal medium. The biosynthesis of PCA relies on the phenazine gene cluster *phzABCDEFG*, and the amino transferase encoded by the *phzE* gene is responsible for the first step of phenazine synthesis [20]. PhzE transfers the amino group of glutamine to chorismite, so that glutamine is the amino donor of the phenazine ring [21]. The de novo synthesis of glutamine takes NH_4_^+^ and α-ketoglutarate as raw materials. Therefore, sufficient NH_4_^+^ is important for the synthesis of phenazine compounds. The experimental results were also consistent with our expectations. In the low concentration range, the higher the NH_4_^+^ concentration, the higher the PCA yield. When the NH_4_^+^ concentration reached 150mM, the production no longer increased and remained at 758.9 mg/L. On the basis of NH_4_^+^ optimization, we further explored the effect of the iron ion valence state and iron concentration on PCA synthesis. The results indicated that Fe^3+^ was more effective than Fe^2+^ in promoting PCA synthesis in the minimal medium, and the optimal concentration was 20 μM. It has been reported that the production of phenazine compounds by *P. chlororaphis* can be increased by adding Fe^3+^ to a complex medium (i.e., Fe^3+^ exhibits a positive effect on phenazine compound production) [22,23], which is consistent with our results.

Herein, we obtained an optimal general defined medium for phenazine synthesis, which was named as GDM (9.9 g/L (NH_4_)_2_HPO_4_, 5.8 g/L K_2_HPO_4_, 3.7 g/L KH_2_PO_4_, 0.12 g/L MgSO_4_, 20 μM FeCl_3_·6H_2_O). According to the raw material prices from Alibaba.com (https://www.alibaba.com/trade/search?tab=all&searchText=crude+glycerol accessed on 16 August 2023), the cost of the GDM was about 10% of a complex medium (e.g., the KB medium), while the PCA production of strain GP72-ANO using the GDM was the same as that achieved using a complex medium. In general, the GDM exhibited great advantages for industrial application. Furthermore, due to its simplicity and well-defined composition, the GDM could also be used to explore intracellular metabolism (such as metabolic flux analysis in new pathways), as well as to determine the utilization efficiency of cells for different carbon sources (such as screening colonies with high carbon source utilization). Overall, the invention of this minimal medium not only contributes to industrial production but also facilitates further research on the synthesis and regulation of phenazine compounds.

In the previous chapter, we proved that enhancing the glycerol metabolic pathway was more conducive to PCA synthesis, which was consistent with previous reports [11,24]. The strain ANOFK1 (strain ANOΔ*mgsA* with *glpF* and *glpK* gene co-overexpression plasmid) obtained the highest yield at 1288.1 mg/L in the GDM, which was 27.4% higher than without engineering the glycerol metabolic pathway. For the sake of production economy and environmental protection, we first attempted to synthesize phenazine compounds using crude glycerol, the by-product of biodiesel production, and the yield was almost the same as with pure glycerol. The subtle differences may have been due to various inhibitors, such as salt, methanol, and heavy metals, present in crude glycerol, which might have inhibited cellular metabolism [25]. Hence, further exploration is needed to overcome these inhibitors in crude glycerol. In a word, this lays the foundation for the use of economical and green carbon sources in the GDM, such as lignocellulose hydrolysates and sugarcane trash.

## 5. Conclusions

Herein, we selected a minimal medium suitable for *P. chlororaphis* to synthesize phenazine compounds. We found that Fe^3+^ and NH_4_^+^ promoted the synthesis of phenazine compounds by *P. chlororaphis*. After component simplification and content optimization, we named this medium as a cost-effective GDM. In general, the cost of the GDM was 1/10 of a complex medium with the same PCA production. In addition, engineering the glycerol metabolic pathway increased the PCA production of the strain by 17.4%, resulting in a value of 1288.1 mg/L, which represents the highest PCA titer achieved by using a cost-effective minimal medium at present.

## Figures and Tables

**Figure 1 biology-12-01292-f001:**
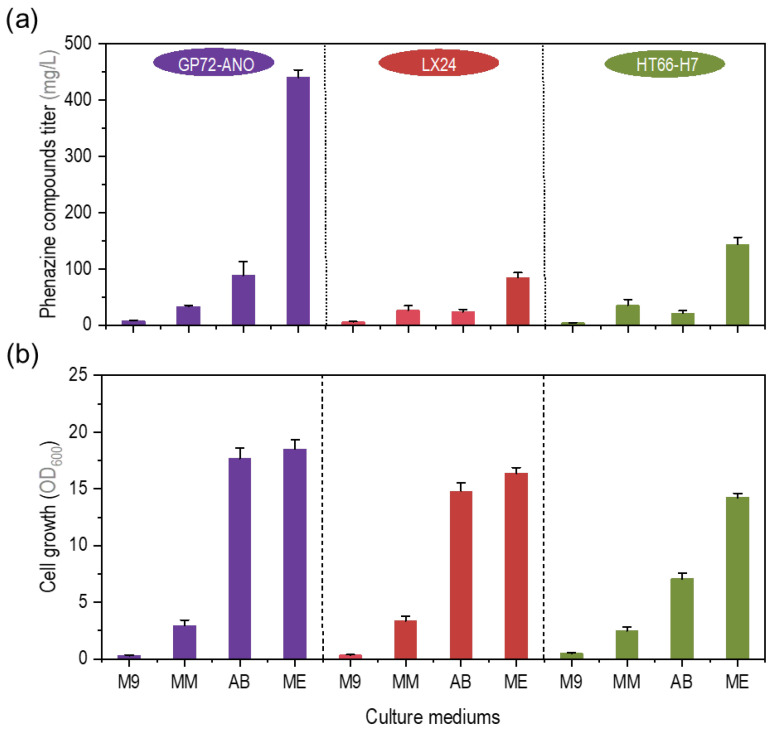
Comparison of fermentation performance of *P. chlororaphis* GP72-ANO (purple), *P. chlororaphis* LX24 (red), and *P. chlororaphis* HT66-H7 (green) in different minimal medium. (**a**) Phenazine compound titers of three strains in four minimal medium. (**b**) Cell growth (optical density at 600 nm, OD_600_) of three strains in four minimal medium. The data represent the means ± SD for three independent cultures.

**Figure 2 biology-12-01292-f002:**
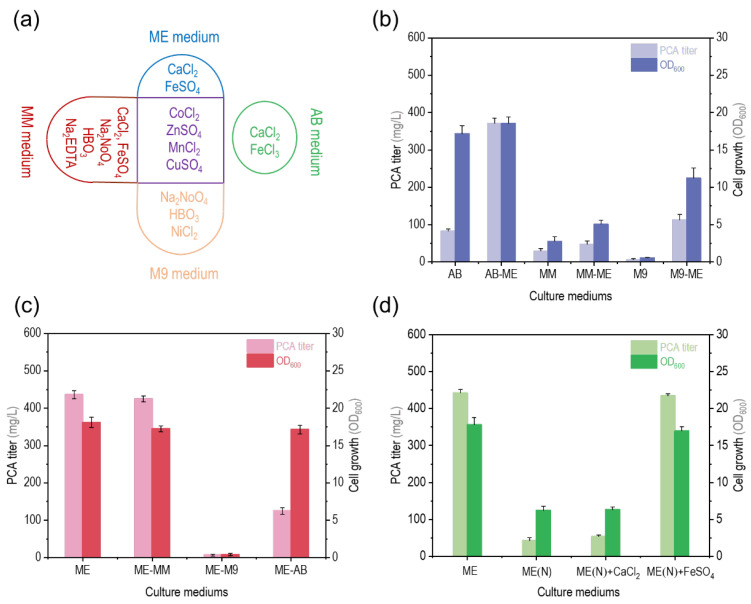
Effect of trace elements in ME medium on PCA titer. (**a**) Differences in trace elements of four minimal medium (ME, AB, MM, and M9). The purple area represents the components common to the trace elements of the ME, MM, and M9 medium. The areas with other colors represent components unique to the trace elements of each medium. (**b**) The PCA titer and cell growth (optical density at 600 nm, OD_600_) of strain *P. chlororaphis* GP72-ANO in the AB; AB-ME (i.e., AB medium with the trace elements of the ME medium instead of the trace elements of the AB medium); MM; MM-ME M9; and M9-ME medium. (**c**) The PCA titer and cell growth (OD_600_) of strain *P. chlororaphis* GP72-ANO in the ME, ME-MM, ME-M9, and ME-AB medium. ME-MM/M9/AB medium: ME medium with the trace elements of the MM/M9/AB medium instead of the trace elements of the ME medium. (**d**) The PCA titer and cell growth (OD_600_) of strain *P. chlororaphis* GP72-ANO in the ME; ME(N) (i.e., ME medium without trace elements); ME(N)+CaCl_2_ (i.e., ME(N) medium with the same CaCl_2_ concentration as the ME medium); and ME(N)+FeSO_4_ (i.e., ME(N) medium with the same FeSO_4_ concentration as the ME medium) medium. The data represent the means ± SD for three independent cultures.

**Figure 3 biology-12-01292-f003:**
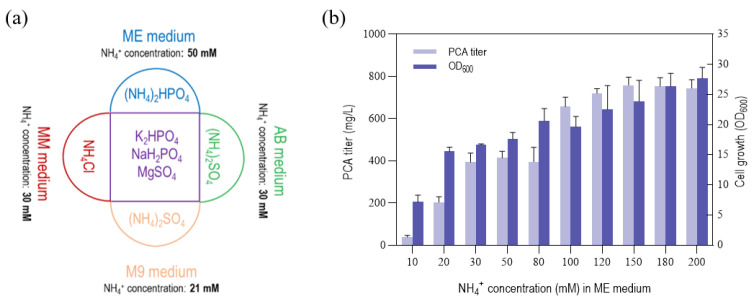
Effects of NH_4_^+^ in ME medium. (**a**) Differences in NH_4_^+^ concentration of four minimal medium (ME, AB, MM, and M9). The purple area represents the components common to the major elements of the ME, MM, and M9 medium. The areas with other colors represent components unique to the major elements of each medium. (**b**) The PCA titer and cell growth (optical density at 600 nm, OD_600_) of *P. chlororaphis* GP72-ANO under different NH_4_^+^ concentrations in the ME(N) + FeSO_4_ (10 μM) medium. The data represent the means ± SD for three independent cultures.

**Figure 4 biology-12-01292-f004:**
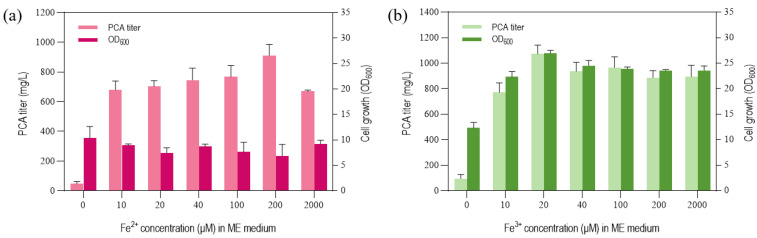
Effects of the addition of different concentrations of Fe^2+^ (**a**) and Fe^3+^ (**b**) on PCA production and cell growth (optical density at 600 nm, OD_600_) of *P. chlororaphis* GP72-ANO cultured in ME(N) medium. The data represent the means ± SD for three independent cultures.

**Figure 5 biology-12-01292-f005:**
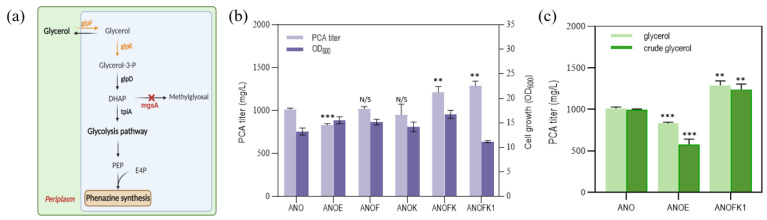
(**a**) Functions of glycerol metabolism genes involved in modifications. (**b**) The PCA titer and cell growth (optical density at 600 nm, OD_600_) in pure glycerol of engineered strains ANO, ANOE (strain ANO with empty plasmid), ANOF (strain ANO with *glpF* gene overexpression plasmid), ANOK (strain ANO with *glpK* gene overexpression plasmid), ANOFK (strain ANO with *glpF* and *glpK* gene co-overexpression plasmid), and ANOFK1 (strain ANOΔ*mgsA* with *glpF* and *glpK* gene co-overexpression plasmid). (**c**) The differences in the PCA titer and cell growth (optical density at 600 nm, OD_600_) in pure glycerol and crude glycerol of engineered strains ANO, ANOE, and ANOFK1. Abbreviations are as follows: DHAP, dihydroxyacetone phosphate; PEP, phosphoenolpyruvate; E4P, erythrose-4-phosphat. The data represent the means ± SD for three independent cultures. Note: ** *p* < 0.01, *** *p* < 0.001; N/S, not significant.

## Data Availability

The data presented in this study are available from the authors.

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
