# Peer review of "Economical Production of Phenazine-1-carboxylic Acid from Glycerol by Pseudomonas chlororaphis Using Cost-Effective Minimal Medium"

_biology, 2023, doi:10.3390/biology12101292_

Round 1
Reviewer 1 Report
In this paper, Zhang and co-workers report a cost-effective minimal medium for the efficient synthesis of phenazine compounds by Pseudomonas chlororaphis. Here some corrections and suggestions to take into consideration.
- It would be helpful to include the biosynthetic pathway in the introduction, as in lines 282-290, to provide a deeper understanding of the parameters evaluated in this work.
- In the results section, various trace elements such as CoCl2, ZnSO4, etc., are mentioned in the different media. However, while attention is drawn to CaCl2 and FeSO4, there is no mention of prior research explaining why these specific compounds are directly related to PCA synthesis. It would be beneficial to delve further into their significance and provide references if there are reports on this relationship.
- In section 3.3, it is necessary to elaborate on the importance of NH4+ and describe the assays conducted to determine the optimal NH4+ concentration.
- Why is 150mM the optimal NH4+ concentration? Isn't the OD600 better at 180mM while maintaining the same PCA concentration obtained?
- Please review the description of Fig. 5.
- Revise the sentence in lines 280-282: " By comparing the differences in NH4+ concentration between four minimal media, we speculated that the higher NH4+ concentration, the higher the yield of phenazines," specifying the optimal NH4+ concentration.
- In lines 301-302, " In general, the cost of GDM medium is 301 1/10 of that of complex medium for the same PCA production" appears redundant.
- In line 312, include the percentage that the highest yield represents to facilitate comparison.
Author Response
- It would be helpful to include the biosynthetic pathway in the introduction, as in lines 282-290, to provide a deeper understanding of the parameters evaluated in this work.
Author response: Thank you for your comment. We added the synthesis pathway of PCA and highlighted the importance of ammonium in the pathway in the introduction section.
- In the results section, various trace elements such as CaCl2, ZnS04, etc., are mentioned in the different media. However, while attention is drawn to CaCl2 and FeS04, there is no mention of prior research explaining why these specific compounds are directly related to PCA synthesis. It would be beneficial to delve further into their significance and provide references if there are reports on this relationship.
Author response: Thank you for your comment. We explained the relationship between these two components and the synthesis of phenazine in the discussion section, and previous references has also been provided and discussed in this section.
- In section 3.3, it is necessary to elaborate on the importance of NH4+ and describe the assays conducted to determine the optimal NH4+ concentration.
Author response: Thank you for your comment. We have added the importance of NH4+ for phenazine compounds in section 3.3, as well as the experimental operation on how to obtain the optimal NH4+ concentration
- Why is 150mM the optimal NH4+ concentration? Isn't the OD600 better at 180mM while maintaining the same PCA concentration obtained?
Author response: Thank you for your comment. We set PCA production as our established goal, so the NH4+ concentration at the highest PCA production is the optimal concentration.
- Please review the description of Fig. 5.
Author response: Thank you for your comment. We apologize for the mistake and we have corrected the description of Fig. 5.
- Revise the sentence in lines 280-282: " By comparing the differences in NH4+ concentration between four minimal media, we speculated that the higher NH4+ concentration, the higher the yield of phenazines," specifying the optimal NH4+ concentration.
Author response: Thank you for your comment. We have removed this sentence and added our speculation about NH4+ may act as a key element to phenazine synthesis and cell growth.
- In lines 301-302, " In general, the cost of GDM medium is 301 1/10 of that of complex medium for the same PCA production" appears redundant.
Author response: Thank you for your comment. We have deleted this sentence.
- In line 312, include the percentage that the highest yield represents to facilitate comparison.
Author response: Thank you for your comment. We have added a description of the degree of increase in production.
Reviewer 2 Report
Major comments:
1- All methods should have a reference, then, indicate the modifications in the protocols
2- There is no statistical analysis, authors should do statistical analysis with one-way or two-way ANOVA for data in figures 1,2,3,4 and 5 to confirm which treatments are significantly different from others. Then, all of df, F, and P should be introduced in the text.
Minor comments
- The sentence of lines 62,63 [and the culture condition of Methanosarcina is harsh and genetic manipulation of Methanosarcina is difficult], it needs to rewritten to be clear
- Lines 92-94: these are results, thus, remove them to be in conclusion
- Delete any references in Results section as in lines 149, 238 with the information mentioned in these paragraphs related to these references and could be transferred to Discussion section.
Minor editing of English language required
Author Response
- All methods should have a reference, then, indicate the modifications in the protocols.
Author response: Thank you for your comment. We have added references to the method.
- There is no statistical analysis, authors should do statistical analysis with one-way or two-way ANOVA for data in figures 1,2,3,4 and 5 to confirm which treatments are significantly different from others. Then, all of df, F, and P should be introduced in the text.
Author response: Thank you for your comment. We have conducted a p-values annotation on Figure 5. Since the p-values in Figure 1,2,3,4 are all less than 0.0001, we did not add any annotations.
-The sentence of lines 62.63 land the culture condition of Methanosarcina is harsh and genetic manipulation of Methanosarcina is difficult], it needs to rewritten to be clear.
Author response: Thank you for your comment. We have expanded and improved this sentence.
-Lines 92-94: these are results, thus, remove them to be in conclusion.
Author response: Thank you for your comment. We have made the modifications according to your suggestion.
-Delete any references in Results section as in lines 149, 238 with the information mentioned in these paragraphs related to these references and could be transferred to Discussion section.
Author response: Thank you for your comment. We have made the modifications according to your suggestion.
Round 2
Reviewer 2 Report
Dear authors
All my comments have been done.
Thanks